# Approximate Leave–One–Out Cross Validation for Robust Scatter Matrix Estimation

**Karim Abou-Moustafa**[*]
Intel Foundry, TDA Research & Development
Intel Corporation
Chandler, AZ 85226

## Abstract

Tyler's $M$-estimator (TME) is an accurate and efficient robust estimator for the scatter matrix when the data are samples from an *elliptical distribution* with heavy-tails and the number of samples $n$ is larger than the number of variables $p$. Unfortunately, when $p > n$, TME is not defined, and various research works have proposed regularized versions of TME using the spirit of Ledoit & Wolf estimator whose performance depends on a carefully chosen *shrinkage coefficient* parameter $\alpha \in (0, 1)$. In this paper, we consider the problem of estimating an optimal shrinkage coefficient $\alpha \in (0, 1)$ for Regularized TME (RTME). In particular, we propose to estimate an optimal shrinkage coefficient by setting $\alpha$ as the solution to a suitably chosen objective function; namely the leave-one-out cross-validated (LOOCV) log-likelihood loss. Since LOOCV is computationally prohibitive even for moderate values of $n$, we propose a computationally efficient approximation for the LOOCV log-likelihood loss that eliminates the need for invoking the RTME procedure $n$ times for each sample left out during the LOOCV procedure. This approximation yields an $O(n)$ reduction in the running time complexity for the LOOCV procedure, which results in a significant speedup for computing the LOOCV estimate. We demonstrate the efficacy of the proposed approach on synthetic high-dimensional data sampled from heavy-tailed elliptical distributions, as well as on real high-dimensional datasets for object and face recognition. Our experiments show that the proposed method is efficient and consistently more accurate than other methods in the literature for shrinkage coefficient estimation.

## 1 Introduction

Covariance matrices, or their scaled versions *scatter matrices*, are ubiquitous in statistical models and procedures for machine learning, pattern recognition, signal processing, and various other fields of science and engineering. The performance of procedures such as principal component analysis (PCA) and its extensions [24], linear discriminant analysis (LDA) and its extensions [31], canonical correlation analysis (CCA) [21], portfolio optimization for investment diversification [28], and outlier detection using robust Mahalanobis distance [4], all depend on an accurate estimate of the covariance matrix. Unfortunately, the process of accurately estimating a covariance matrix is challenging since the number of unknown parameters grows quadratically with the number of variables (or features) $p$. The problem is well-understood when the number of samples $n$ is much larger than $p$ and the data's underlying distribution is a multivariate Gaussian. In this case, the sample covariance matrix (SCM) is an accurate estimate of the covariance matrix, and is optimal under most criteria [43].

---

[*]`karim.abou-moustafa@intel.com, karim.aboumoustafa@gmail.com`

39th Conference on Neural Information Processing Systems (NeurIPS 2025) Workshop: Reliable ML from Unreliable Data.

In various modern applications, however, $p$ may be comparable to, or greater than $n$, and the data's underlying distribution may be non-Gaussian and/or *heavy-tailed*. The situation gets exacerbated if the data are also contaminated with outliers. In such settings, the SCM is known to be a poor estimate of the covariance matrix and one needs to consider estimators that are more accurate and robust than the SCM. In this work, we are interested in a particular estimator from the family of *robust* and *affine-invariant* $M$-estimators of scatter matrices proposed by Maronna [29] – namely Tyler's $M$-estimator (TME) [42, 41] – in the setting where the data's distribution is *heavy-tailed* and the sample support is relatively low; i.e. $p$ is large, and $p \geq n$.[2]

Various approaches were proposed for estimating high-dimensional covariance matrices when $p \geq n$; shrinkage-based approaches [23, 11, 9, 27]; specifying an appropriate prior distribution for the covariance matrix [17]; regularization-based approaches [10, 39, 38]; approaches that exploit sparsity assumptions (banding, tapering, thresholding) [3, 25, 5, 19]; and approaches developed in the robust statistics literature [22, 18]. Except for some approaches from the robust statistics literature, most of the other approaches assume that the data's underlying distribution is a multivariate Gaussian, which may not be a reasonable assumption for handling outliers, or samples from heavy-tailed distributions.

TME is an accurate and efficient robust estimator for the scatter matrix when the data are samples from an *elliptical distribution* with heavy-tails and $n \gg p$. Elliptical distributions (introduced shortly) are the generalization of the multivariate Gaussian distribution and are suitable for modeling empirical distributions with heavy tails, where such heavy tails may be due to the existence of outliers in the data [32]. In this setting, and under some mild assumptions on the data, TME has various attractive properties [42, 41]. TME is strongly consistent, asymptotically normal, and is the *most robust* estimator for the scatter matrix for an elliptical distribution in a *minimax* sense; minimizing the maximum asymptotic variance (see Remark 3.1 in [42]). Unfortunately, in the $p > n$ regime, TME is not defined. Various research works have proposed regularized versions of TME using the spirit of Ledoit–Wolf estimator [27] whose performance depends on a *carefully chosen* regularization parameter, or *shrinkage coefficient* $\alpha \in (0, 1)$ [1, 7, 43, 37, 40, 35, 45, 2]. Our work here addresses the question of shrinkage coefficient estimation for *Regularized* TME (RTME), and proposes a computationally efficient algorithm for obtaining a near-optimal estimate for this parameter.[3]

Unfortunately, the recursive nature of TME's procedure makes estimating an optimal shrinkage coefficient for this estimator a non-trivial problem. Arguably, three broad approaches were considered to address this problem: (*i*) oracle and random matrix theory (RMT) based approaches [7, 35, 8, 45, 20]; (*ii*) data-dependent approaches based on *Cross Validation* (CV) techniques [1, 43, 40, 12]; and (*iii*) maximum likelihood (ML) based approaches [2]. Oracle-based approaches are computationally efficient due to their closed-form solutions but may come short in terms of accuracy due to their implicit assumptions on the data distribution, and due to the implicit assumptions in their asymptotic estimates. CV techniques on the other hand are more accurate than oracle based methods since they are data-dependent approaches; this accuracy, however, comes at the cost of intensive computations, especially for high-dimensional data, which makes CV techniques not a favorable option for various applications. [2] proposed a ML based approach, namely the expected likelihood (EL) method, for selecting a shrinkage coefficient for RTME when used for some specific problems in wireless communications; *e.g.* adaptive-filtering and estimating the signal's direction of arrival. While in such applications the noisy data samples may be reasonably assumed to have an elliptical distribution, the EL method may not be considered a general approach for estimating the shrinkage coefficient due to the special controlled environments for such problems in wireless communications.

In this work, we propose a more general approach for estimating an optimal shrinkage coefficient $\alpha^*$ for RTME. Our proposed approach formulates the problem of finding an optimal shrinkage coefficient as an optimization problem with respect to parameter $\alpha$. In particular, we define an optimal shrinkage coefficient $\alpha^*$ as the minimizer for the following loss function; the leave-one-out cross-validated (LOOCV) negative log-likelihood (NLL) for the estimated scatter matrix with respect to parameter $\alpha$ (Eq. 12). Since LOOCV is computationally prohibitive even for moderate values of $n$, we propose a computationally efficient *approximation* for the LOOCV NLL loss that *eliminates* the need for computing the Regularized TME $n$ times for each sample left out during the LOOCV procedure. This

_______________

[2]See [30, 44] for a recent overview and results on this family of estimators.

[3]Shrinkage coefficient estimation for SCM, and *generalized* $M$-estimators for elliptically distributed data, were considered in [34, 36].

approximation yields an $O(n)$ reduction in the running time complexity for the LOOCV procedure, which results in a significant speedup in computing the LOOCV NLL loss.

At high-level, the resulting procedure, the *Approximate Cross-Validated Likelihood* (ACVL) method, exploits mild computation and the given finite sample to select a (*data-dependent*) *near-optimal* coefficient $\alpha$ for RTME. In addition, the ACVL method is amenable to parallel computation, and is directly applicable to sparse covariance matrix estimation by means of thresholding the Regularized TME [15]. We demonstrate the efficiency and accuracy of the ACVL method on synthetic high-dimensional data sampled from heavy-tailed elliptical distributions, as well as on real high-dimensional datasets for face recognition (Yale B), object recognition (CIFAR10 and CIFAR 100), and handwritten digit recognition (USPS). Our experiments show that, with some additional mild computation, the ACVL method is efficient and more accurate than other methods in the literature.

## 1.1 Notation and Setup

Scalars and indices are denoted by lowercase letters: $x, y$ and $i, j$, respectively. Vectors are denoted by lowercase bold letters: $\mathbf{x}, \mathbf{y}$, and matrices by uppercase bold letters: $\mathbf{X}, \mathbf{Y}$. Sets are denoted by calligraphic letters: $\mathcal{X}, \mathcal{Y}$, and spaces are denoted by double-bold uppercase letters: $\mathbb{R}, \mathbb{S}$. The identity matrix is denoted by $\mathbf{I}$, and $\mathbf{0}$ is the vector with all zeros, both with suitable dimensions from the context. For $\mathbf{x} \in \mathbb{R}^p$, $\|x\|$ is the Euclidean norm. For a matrix $\mathbf{A} = (a_{ij})$, $\|\mathbf{A}\|_F$ is the Frobenius norm, $\mathrm{Tr}(\mathbf{A})$ is the matrix trace, and $\det(\mathbf{A})$ is the matrix determinant. The space of symmetric and positive definite (PD) matrices is denoted by $\mathbb{S}_+^p$. The unit sphere in $\mathbb{R}^p$ is denoted by $\mathcal{S}^p$, where $\mathcal{S}^p = \{\mathbf{x} \in \mathbb{R}^p \ s.t. \ \|\mathbf{x}\| = 1\}$.

## 1.2 Elliptical Distributions

We will use the stochastic model due to [6] and recently used in the literature to define *elliptical* random vectors (RV) [13, 15]. Let $\mathbf{z}$ be a $p$ dimensional RV generated by the following model:

$$\mathbf{z} = \boldsymbol{\mu} + u\mathbf{S}^{\frac{1}{2}}\mathbf{y} = \boldsymbol{\mu} + u\tilde{\mathbf{x}}, \qquad (1)$$

where $\boldsymbol{\mu} \in \mathbb{R}^p$ is a location vector, $\mathbf{S} \in \mathbb{S}_+^p$ is a *scatter* or *shape* matrix, $\mathbf{y}$ is drawn uniformly from $\mathcal{S}^p$, and $u$ is a nonnegative random variable (r.v.) stochastically independent of $\mathbf{y}$. The resulting RV $\mathbf{z}$ from the model in (1) is an *Elliptically Distributed* (ED) RV. Note that $\mathbf{S}$ in (1) is not unique since it can be arbitrarily scaled with $1/u$ absorbing the scaling factor $u$. The distribution function of $u$, known as the *generating distribution function*, constitutes the particular elliptical distribution family of the RV $\mathbf{z}$. If $\mathbf{z}$ is an ED RV, its probability density function (PDF) is defined as:

$$f(\mathbf{z}; \boldsymbol{\mu}, \mathbf{S}, g_u) = \det(\mathbf{S})^{-\frac{1}{2}} g_u \left( \bar{\mathbf{z}}^\top \mathbf{S}^{-1} \bar{\mathbf{z}} \right), \qquad (2)$$

where $\bar{\mathbf{z}} = (\mathbf{z} - \boldsymbol{\mu})$, and $g_u : \mathbb{R}_+ \mapsto \mathbb{R}_+$ is a nonnegative decreasing function known as the *density generator function* and is not dependent on $\boldsymbol{\mu}$ and $\mathbf{S}$, but dependent on the generating distribution function of $u$. The density generator function determines the shape of the PDF, as well as the *tail decay* of the distribution. For any elliptical distribution, if its population covariance matrix $\boldsymbol{\Sigma}$ exists, then $\boldsymbol{\Sigma} = c_g \mathbf{S}$ for some constant $c_g > 0$ that is dependent on $g_u$.

## 2 Regularized Tyler's M–Estimator

Let $\mathcal{Z}_n = (\mathbf{z}_1, \ldots, \mathbf{z}_n)$ be a sample of $n$ *independent* and *identically distributed* (*i.i.d.*) realizations from the model in (1) with location vector $\boldsymbol{\mu} = \mathbf{0}$ and scatter matrix $\mathbf{S}$. We are interested in computationally efficient and statistically accurate algorithms for estimating the population scatter matrix $\mathbf{S}$ using the samples in $\mathcal{Z}_n$ in the setting where $p > n$. Here we do not make *a priori* sparsity assumptions on the scatter matrix $\mathbf{S}$. Without any *a priori* knowledge on $c_g$ and $g_u$, it may seem less probable to obtain a good estimator for $\mathbf{S}$. In addition, for some elliptical distributions – such as the multivariate Cauchy distribution – they may have infinite second moments in which case the population covariance matrix $\boldsymbol{\Sigma}$ does not exist. Thus it may always be better to consider and estimate the *normalized* scatter matrix $\mathbf{S}$ which is always defined [37].

TME can be derived as a ML estimator of the shape matrix for the *Angular Central Gaussian* (ACG) distribution (defined in Equation 3) based on the sample $\mathcal{Z}_n$ [41]. With $\boldsymbol{\mu} = \mathbf{0}$, the sample $\mathcal{Z}_n$ can be

written as $(u_1 \tilde{\mathbf{x}}_1, \ldots, u_n \tilde{\mathbf{x}}_n)$. Since the scalars $u_1, \ldots, u_n$ are unknown, there is a scaling ambiguity and one can only expect to estimate matrix $\mathbf{S}$ up to a scaling factor. TME overcomes this limitation by working with the normalized samples: $\mathbf{x}_i = \mathbf{z}_i / \|\mathbf{z}_i\| = \tilde{\mathbf{x}}_i / \|\tilde{\mathbf{x}}_i\|$, $1 \leq i \leq n$, where the scalars $u_i$ cancels out. The PDF for the vectors $\mathbf{x}_1, \ldots, \mathbf{x}_n$ is given by:

$$f(\mathbf{x}; \mathbf{S}) = (2\pi)^{-\frac{p}{2}} \Gamma(\tfrac{1}{2}) \det(\mathbf{S})^{-\frac{1}{2}} \left(\mathbf{x}^\top \mathbf{S}^{-1} \mathbf{x}\right)^{-\frac{p}{2}}, \tag{3}$$

where $\mathbf{x} \in \mathcal{S}^p$, $\Gamma(\cdot)$ is the Gamma function, and $\Gamma(p/2)/(2\pi)^{\frac{p}{2}}$ is the surface area of $\mathcal{S}^p$. The ACG density in (3) represents the *distribution of directions* for samples drawn from a multivariate Gaussian distribution with zero mean and covariance matrix $\mathbf{S}$ [41]. Thus, only the directions of outliers can affect TME's performance but not their magnitude. Given an *i.i.d.* random sample $\mathcal{X}_n = (\mathbf{x}_1, \ldots, \mathbf{x}_n)$ from a distribution having the ACG density in (3), the likelihood of $\mathcal{X}_n$ with respect to $\mathbf{S}$ is *proportional* to:

$$L(\mathcal{X}_n; \mathbf{S}) = \det(\mathbf{S})^{-n/2} \prod_{i=1}^{n} \left(\mathbf{x}_i^\top \mathbf{S}^{-1} \mathbf{x}_i\right)^{-\frac{p}{2}}. \tag{4}$$

Taking $-\log$ of $L(\mathcal{X}_n; \mathbf{S})$ yields the loss function:

$$\mathcal{L}(\mathcal{X}_n; \mathbf{S}) = \frac{p}{2} \sum_{i=1}^{n} \log\left(\mathbf{x}_i^\top \mathbf{S}^{-1} \mathbf{x}_i\right) + \frac{n}{2} \log \det(\mathbf{S}), \tag{5}$$

which will be needed for our next discussions. Taking the derivative of $\mathcal{L}(\mathcal{X}_n; \mathbf{S})$ with respect to $\mathbf{S}$ and equating it to zero, the ML estimator for $\mathbf{S}$ is the solution to the following fixed point equation:

$$\mathbf{S}_n = \frac{p}{n} \sum_{i=1}^{n} \mathbf{x}_i \mathbf{x}_i^\top / (\mathbf{x}_i^\top \mathbf{S}_n^{-1} \mathbf{x}_i), \tag{6}$$

where $\mathbf{x}_i \neq \mathbf{0}$, for $i = 1, \ldots, n$ since samples lying at the origin provide no directional information on $\mathbf{S}$. If $n > p(p-1)$, Theorem 1 in [41] states that with probability one, the ML estimate of $\mathbf{S}$ *exists*, corresponds to the solution in (6), and is *unique* up to a positive multiplicative scalar. The solution to (6) can be found using the following fixed point iteration (FPI) algorithm:

$$\widehat{\mathbf{S}}_{t+1} = \frac{p}{n} \sum_{i=1}^{n} \mathbf{x}_i \mathbf{x}_i^\top / (\mathbf{x}_i^\top \widehat{\mathbf{S}}_t^{-1} \mathbf{x}_i), \tag{7}$$

with $\widehat{\mathbf{S}}_0 = \mathbf{I}$, or any arbitrary initial $\widehat{\mathbf{S}}_0 \in \mathbb{S}_+^p$ [26]. Theorem 2.2 and Corollaries 2.2 & 2.3 in [42] show that if $n > p + 1$ and assuming that every $p$ samples out of $\mathcal{X}_n$ are *linearly independent* with probability one, and that the maximum likelihood estimate of $\mathbf{S}$ exists, then the FPI algorithm in (7) *almost surely* converges to the solution of (6), and the limiting solution $\widehat{\mathbf{S}} = \widehat{\mathbf{S}}_T$ computed at the last iterate $T$ is unique up to a positive multiplicative scalar.

TME has various attractive properties and is asymptotically optimal under different criteria. In particular, TME is strongly consistent, asymptotically normal, and is the *most robust* estimator for the scatter matrix for an elliptical distribution in a *minimax* sense; minimizing the maximum asymptotic variance (see Remark 3.1 in [42]). Unfortunately, when $p > n$, TME is not defined; the LHS of (6) must be a full rank symmetric PD matrix, while the RHS is rank-deficient.[4] Various researchers have proposed different flavors of RTME using the spirit of [27] linear shrinkage estimator [1, 7, 43, 37, 35]. In particular, Sun *et al.* (SBP) [40] proposed the following penalized log-likelihood function to derive a regularized version of TME:

$$\mathcal{L}_\mathcal{P}(\mathcal{X}_n; \mathbf{S}) = \mathcal{L}(\mathcal{X}_n; \mathbf{S}) + \beta \mathcal{P}(\mathbf{S}), \tag{8}$$

where $\mathcal{L}(\mathcal{X}_n; \mathbf{S})$ is defined in (5), and $\mathcal{P}(\mathbf{S})$ is a penalty function defined as: $\mathcal{P}(\mathbf{S}) = \text{Tr}(\mathbf{S}^{-1}\mathbf{T}) + \log \det(\mathbf{S})$, with $\beta > 0$ is the regularization parameter (or shrinkage coefficient). Matrix $\mathbf{T} \in \mathbb{S}_p^+$ is a given target matrix with some desirable structural properties (diagonal, banded, Toeplitz, etc.). Letting $\alpha = \beta/(1+\beta)$, the solution to (8) has to satisfy the fixed point equation:

$$\mathbf{S}_n = (1 - \alpha)\frac{p}{n} \sum_{i=1}^{n} \frac{\mathbf{x}_i \mathbf{x}_i^\top}{\mathbf{x}_i^\top \mathbf{S}_n^{-1} \mathbf{x}_i} + \alpha \mathbf{T}. \tag{9}$$

---

[4]Regularization may still be needed for $p \leq n \leq p(p-1)$ when the points are not in general position, and/or the samples are not drawn from an elliptical distribution.

Note that $\alpha \in (0, 1)$ for any $0 < \beta < \infty$. Starting from an arbitrary $\widehat{\mathbf{S}}_0 \in \mathbb{S}_p^+$, the final solution can be obtained using the *Regularized* FPI (RFPI) algorithm:

$$\widehat{\mathbf{S}}_{t+1}(\alpha) = (1 - \alpha)\frac{p}{n}\sum_{i=1}^{n}\frac{\mathbf{x}_i\mathbf{x}_i^{\top}}{\mathbf{x}_i^{\top}\widehat{\mathbf{S}}_t^{-1}(\alpha)\mathbf{x}_i} + \alpha\mathbf{T} , \tag{10}$$

where $\alpha \in (0, 1)$ is the *shrinkage coefficient* that controls the amount of shrinkage applied to scatter matrix $\mathbf{S}$ towards the target matrix $\mathbf{T}$. Theorem 11 and Proposition 13 in [40] establish the necessary and sufficient conditions for the *existence* and *uniqueness* of the solution to Equation (9), while Proposition 18 ensures that the RFPI in (10) converges to the unique solution of (9).

Without loss of generality, if $\mathbf{T} = \mathbf{I}$ and $\alpha = 0$, one restores the unbiased TME in (7), and if $\alpha = 1$ the estimator reduces to the uncorrelated scatter matrix $\alpha\mathbf{I}$. If $p < n$, and the samples are drawn from an elliptical distribution, $\alpha$ is expected to be zero (or close to zero) and results for existence and uniqueness of the estimator still hold [37]. If $p \geq n$, $\alpha$ is expected to be large[5]; however to ensure the *existence and uniqueness* of the estimator, $\alpha$ needs to be strictly greater than $1 - n/p$ [37, 40].

## 2.1 Runtime Analysis for the RFPI Algorithm

The magnitude of $\alpha$ has an impact on the accuracy of the final estimate $\widehat{\mathbf{S}} = \widehat{\mathbf{S}}_T$, as well as on the convergence speed for the RFPI algorithm. In particular, Lemma 1 in [15] gives a result on the *uniform linear convergence* of the algorithm in (10) to a unique solution; for desired accuracy $\varepsilon$, convergence ratio $r$, and sufficiently large $\alpha > p/n - 1$, *at most* $\lceil\log_{1/r}(1/\varepsilon)\rceil$ iterations are needed for (10) to converge to the unique solution of (9). A preliminary analysis of the RFPI algorithm shows that the running time for each iteration is $O(np^2 + p^3)$ where $O(np^2)$ is the time needed to compute the sum of rank-one matrices, and $O(p^3)$ is the time needed to compute the inverse matrix $\widehat{\mathbf{S}}_t^{-1}(\alpha)$. Since $\widehat{\mathbf{S}}_t(\alpha)$ is PD, an efficient computation for the inverse can be done using Cholesky factorization [16]: $\widehat{\mathbf{S}}_t(\alpha) = \mathbf{L}\mathbf{L}^{\top}$, where $\mathbf{L}$ is a lower triangular matrix. Cholesky factorization requires $\frac{1}{3}p^3$ flops: $\frac{1}{6}p^3$ multiplications, and $\frac{1}{6}p^3$ additions. Finally inverting a triangular matrix will require $p^2$ flops. If $T$ iterations are needed for the RFPI algorithm to converge, its total running time complexity will be $O(T(np^2 + p^3))$.

## 3 Optimal Choice of Shrinkage Coefficient $\alpha$

Our objective is to find an appropriate $\alpha$ that is *optimal* under a suitable loss function. If the true scatter matrix $\mathbf{S}$ is known, one can choose a shrinkage coefficient that minimizes an appropriate distance metric between $\widehat{\mathbf{S}}$ and $\mathbf{S}$. Since $\mathbf{S}$ is unknown, our approach will depend on the loss function of $\mathcal{X}_n$ with respect to $\mathbf{S}$ in (5). In particular, for a *fixed* $\bar{\alpha} \in (0, 1)$, suppose that $\widehat{\mathbf{S}}(\bar{\alpha})$ is an estimate of the true scatter matrix $\mathbf{S}$. Given the sample $\mathcal{X}_n$, one can assess the quality of $\widehat{\mathbf{S}}(\bar{\alpha})$ with respect to $\mathcal{X}_n$ using the loss function $\mathcal{L}(\mathcal{X}_n; \mathbf{S})$ in (5), by replacing $\mathbf{S}$ with $\widehat{\mathbf{S}}(\bar{\alpha})$. Using this approach, an optimal $\alpha$ with respect to $\mathcal{X}_n$, denoted $\alpha^*$, will be the one that minimizes $\mathcal{L}(\mathcal{X}_n, \widehat{\mathbf{S}}(\alpha))$ over $\alpha \in (0, 1)$; i.e.

$$\alpha^* = \underset{\alpha \in (0,1)}{\arg\min} \quad \mathcal{L}(\mathcal{X}_n; \widehat{\mathbf{S}}(\alpha)) . \tag{11}$$

The problem with this direct approach is that $\widehat{\mathbf{S}}(\alpha)$ needs to be computed using the sample $\mathcal{X}_n$. That is, $\mathcal{X}_n$ will be used twice; first time to compute $\widehat{\mathbf{S}}(\alpha)$, and a second time to assess the quality of $\widehat{\mathbf{S}}(\alpha)$ using $\mathcal{L}(\mathcal{X}_n; \widehat{\mathbf{S}}(\alpha))$ in (5). This is known as *double dipping* and inevitably it leads to an *overfit* estimate of $\alpha$.

CV techniques overcome this problem by splitting the data into two non-overlapping samples; one sample for estimating $\mathbf{S}$ and the other sample for estimating the loss $\mathcal{L}$. Here, we propose to use *Leave-One-Out* CV (LOOCV) for estimating $\mathbf{S}$ and $\mathcal{L}$. In particular, for $1 \leq i \leq n$, LOOCV splits $\mathcal{X}_n$ into two sub-samples: the sample $\mathcal{X}_{n \backslash i} = (\mathbf{x}_1, \ldots, \mathbf{x}_{i-1}, \mathbf{x}_{i+1}, \ldots, \mathbf{x}_n)$, and the sample $(\mathbf{x}_i)$ which contains the single data point $\mathbf{x}_i$. The sample $\mathcal{X}_{n \backslash i}$ will be used to estimate $\mathbf{S}(\alpha)$ using the RFPI

---

[5]If $p < n$ and the samples are heavy-tailed and not from an elliptical distribution, $\alpha$ is expected to be large as well.

algorithm in (10), while the single sample $(\mathbf{x}_i)$ will be used to estimate $\mathcal{L}(\mathbf{x}_i; \widehat{\mathbf{S}}(\alpha))$. This process is repeated $n$ times and the LOOCV estimate will be the average of all $\mathcal{L}(\mathbf{x}_i; \widehat{\mathbf{S}}(\alpha))$, $1 \leq i \leq n$. Using LOOCV, an optimal $\alpha$ can be computed as follows:

$$\widehat{\alpha}_{CV}^* = \underset{\alpha \in (0,1)}{\arg \min} \quad \mathcal{L}_{CV}(\mathcal{X}_n, \alpha) \,, \tag{12}$$

where $\mathcal{L}_{CV}(\cdot)$ is the average CV *Loss* (CVL) defined as:

$$\mathcal{L}_{CV}(\mathcal{X}_n, \alpha) = \frac{1}{n} \sum_{i=1}^{n} \mathcal{L}(\mathbf{x}_i; \widehat{\mathbf{S}}(\alpha; \mathcal{X}_{n\setminus i})) \,, \tag{13}$$

and $\widehat{\mathbf{S}}(\alpha; \mathcal{X}_{n\setminus i})$ is estimated from the points in $\mathcal{X}_{n\setminus i}$ using the RFPI algorithm (10). In practice, one possible approach to solve problem (12) can be using a simple *grid search*: (*i*) define a discrete range of increasing values of $\alpha$: $(\alpha_1, \ldots, \alpha_j, \ldots, \alpha_m)$; (*ii*) evaluate $\mathcal{L}_{CV}(\mathcal{X}_n, \alpha_j)$ for each $\alpha_j$ using (13); and (*iii*) choose $\alpha_j$ with the minimum $\mathcal{L}_{CV}(\cdot)$.[6] For a *fine* discretization for the range of $\alpha$'s, this direct estimation approach will yield an estimate for $\alpha$ that is *reasonably close* to its optimal value. With little abuse of terminology, we refer to this method for estimating $\alpha^*$ as the *Exact CVL* method.

### 3.1 The Computational Overhead of LOOCV

LOOCV is notorious for its high computational overhead. Indeed, for a fixed $\bar{\alpha}$ and for $n$ samples in $\mathcal{X}_n$, LOOCV will make $n$ calls for the RFPI algorithm in order to compute $\mathcal{L}(\mathbf{x}_i, \widehat{\mathbf{S}}(\alpha; \mathcal{X}_{n\setminus i}))$ in the RHS of (13). Thus, for $m$ values of $\alpha_j$ from $(\alpha_1, \ldots, \alpha_m)$, the Exact CVL method in (12) will require $mn$ calls for the RFPI algorithm, which is prohibitive even for moderate values of $n$. If the RFPI algorithm requires $T$ iterations to converge, then the RFPI algorithm will consume $O(mn.T(np^2 + p^3))$ time from the Exact CVL method in (12), where $O(T(np^2 + p^3))$ is the running time for a single call for the RFPI algorithm.

Our objective in the following section is to reduce the time consumed by the RFPI algorithm in the Exact ACVL method by a factor of $n$; i.e. to be $O(m.T(np^2 + p^3))$ instead of $O(mn.T(np^2 + p^3))$. In particular, we propose an efficient approximation for $\widehat{\mathbf{S}}(\alpha, \mathcal{X}_{n\setminus i})$ in (13) so that the RFPI algorithm is invoked $m$ times only instead of $mn$ times to compute $\mathcal{L}_{CV}(\mathcal{X}_n, \alpha)$ in (12). The gain in speed due to this approximation while maintaining the accuracy of the estimated $\alpha$ is depicted in Figure (1) for the multivariate Cauchy distribution. In particular, Figure (1) depicts the *Exact* CVL method vs. the *approximation* developed in the following section in terms of the average CV loss in (13), running time (in seconds), and the optimal $\alpha$ obtained from each method (details in §5).

## 4 Approximate LOOCV

The approximation proposed next is based on rewriting the RFPI algorithm in a more enlightening form. For a fixed $\bar{\alpha}$, the RFPI algorithm in (10) can be expressed as follows:

$$\widehat{\mathbf{S}}_{t+1}(\bar{\alpha}) = (1 - \bar{\alpha})p \left( \frac{1}{n} \sum_{i=1}^{n} w_{t,i}^{-1} \mathbf{x}_i \mathbf{x}_i^\top \right) + \bar{\alpha} \mathbf{T} \,, \tag{14}$$

where $w_{t,i} = \mathbf{x}_i^\top \widehat{\mathbf{S}}_t^{-1}(\bar{\alpha}) \mathbf{x}_i$, and $t = 1, \ldots, T$. That is, the first term in the RHS of (14) involves a weighted sample covariance matrix using the weights $w_{t,i}$ and the RFPI algorithm iteratively estimates these weights until convergence. For initial matrix $\widehat{\mathbf{S}}_0 \in \mathbb{S}_+^p$, let $(\widehat{w}_1, \widehat{w}_2, \ldots, \widehat{w}_n)$ be the optimal weights estimated using $\mathcal{X}_n$ and the RFPI in (14). Then, the *final* estimate for the scatter matrix can be written as:

$$\widehat{\mathbf{S}}(\bar{\alpha}; \mathcal{X}_n) = (1 - \bar{\alpha}) \frac{p}{n} \sum_{i=1}^{n} \frac{1}{\widehat{w}_i} \mathbf{x}_i \mathbf{x}_i^\top + \bar{\alpha} \mathbf{T} \,. \tag{15}$$

---

[6]Note that when $p > n$, and for existence and uniqueness results to hold, $\alpha$ needs to be strictly greater than $1 - n/p$ [37, 40], and hence there is no need to evaluate $\mathcal{L}_{CV}(\cdot)$ for $\alpha \leq 1 - n/p$.

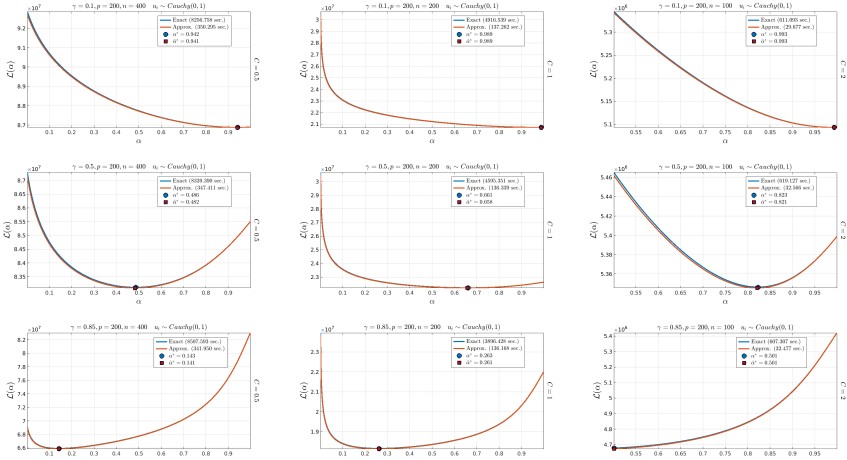

Figure 1: Comparison between *Exact* and *Approximate* CVL for samples drawn from a multivariate Cauchy distribution in three different settings; $p < n$ (left), $p = n$ (middle), and $p > n$ (right), and for three different values of $\gamma = \{0.1, 0.5, 0.85\}$. The blue circle and red square indicate the optimal values for $\alpha$ obtained from the Exact and Approximate CVL methods, respectively. The running times (in seconds) for the Exact and Approximate CVL methods are shown in the legend. The speedup for the Approximate CVL method for each sub-figure is: (first row) 24.3x, 35.8x, 20.6x; (second row) 24.0x, 33.7x, 19.0x; (third row) 25.0x, 28.6x, 18.7x .

Let $\mathcal{X}_{n\setminus i} = (\mathbf{x}_1, \ldots, \mathbf{x}_{i-1}, \mathbf{x}_{i+1}, \ldots, \mathbf{x}_n)$. Similar to (15), using $\bar{\alpha}$ and initial matrix $\widehat{\mathbf{S}}_0$, the *final* estimate $\widehat{\mathbf{S}}$ using $\mathcal{X}_{n\setminus i}$ and the RFPI in (14) will be:

$$\widehat{\mathbf{S}}(\bar{\alpha}; \mathcal{X}_{n\setminus i}) = (1 - \bar{\alpha}) \frac{p}{n-1} \sum_{\substack{j=1 \\ j \neq i}}^{n} \frac{1}{\widehat{v}_j} \mathbf{x}_j \mathbf{x}_j^\top + \bar{\alpha} \mathbf{T} \, , \tag{16}$$

where $(\widehat{v}_1, \ldots, \widehat{v}_{i-1}, \widehat{v}_{i+1}, \ldots, \widehat{v}_n)$ are the optimal weights estimated using $\mathcal{X}_{n\setminus i}$. In terms of computations, and for a fixed $\bar{\alpha} \in (0, 1)$, computing the final estimate $\widehat{\mathbf{S}}(\bar{\alpha}; \mathcal{X}_{n\setminus i})$ for each $i = 1, \ldots, n$ requires invoking the RFPI algorithm $n$ times during the LOOCV procedure. This yields a total running time of $O(nT(np^2 + p^3))$ which is inefficient even for moderate values of $n$ and $p$.

Suppose that the true scatter matrix $\mathbf{S}^* \in \mathbb{S}_p^+$ is known and $(\mathbf{S}^*)^{-1}$ has been computed. The final estimate $\widehat{\mathbf{S}}(\bar{\alpha}; \mathcal{X}_n)$ in (15) can be *directly* computed without invoking the RFPI algorithm in (14):

$$\widehat{\mathbf{S}}(\bar{\alpha}; \mathcal{X}_n) = \frac{(1 - \bar{\alpha})p}{n} \sum_{i=1}^{n} \frac{1}{\widehat{w}_i^*} \mathbf{x}_i \mathbf{x}_i^\top + \bar{\alpha} \mathbf{T} \, , \tag{17}$$

where $\widehat{w}_i^* = \mathbf{x}_i^\top (\mathbf{S}^*)^{-1} \mathbf{x}_i$. Similarly, using $(\mathbf{S}^*)^{-1}$, the final estimate $\widehat{\mathbf{S}}(\bar{\alpha}; \mathcal{X}_{n\setminus i})$ in (16) can be *directly* computed without invoking the RFPI algorithm in (14):

$$\widehat{\mathbf{S}}(\bar{\alpha}; \mathcal{X}_{n\setminus i}) = \frac{(1 - \bar{\alpha})p}{n-1} \sum_{\substack{j=1 \\ j \neq i}}^{n} \frac{1}{\widehat{v}_j^*} \mathbf{x}_j \mathbf{x}_j^\top + \bar{\alpha} \mathbf{T} \, , \tag{18}$$

where $\widehat{v}_j^* = \mathbf{x}_j^\top (\mathbf{S}^*)^{-1} \mathbf{x}_j$. Note that both $\widehat{w}_i^*$ in (17) and $\widehat{v}_j^*$ in (18) are dependent on the true but unknown scatter matrix $\mathbf{S}^*$ and in this case: $\widehat{v}_j^* = \widehat{w}_j^*$ for $j \neq i$, and $j = 1, \ldots, n$. Since $\mathbf{S}^*$ is unknown, we propose to approximate $\widehat{\mathbf{S}}(\bar{\alpha}; \mathcal{X}_{n\setminus i})$ in (18) using the following estimate:

$$\widetilde{\mathbf{S}}(\bar{\alpha}; \mathcal{X}_{n\setminus i}) = \frac{(1 - \bar{\alpha})p}{n-1} \sum_{\substack{j=1 \\ j \neq i}}^{n} \frac{1}{\widetilde{v}_j} \mathbf{x}_j \mathbf{x}_j^\top + \bar{\alpha} \mathbf{T}, \text{ where} \tag{19}$$

$$\widetilde{v}_j = \mathbf{x}_j^\top \widehat{\mathbf{S}}(\bar{\alpha}; \mathcal{X}_n)^{-1} \mathbf{x}_j \, .$$

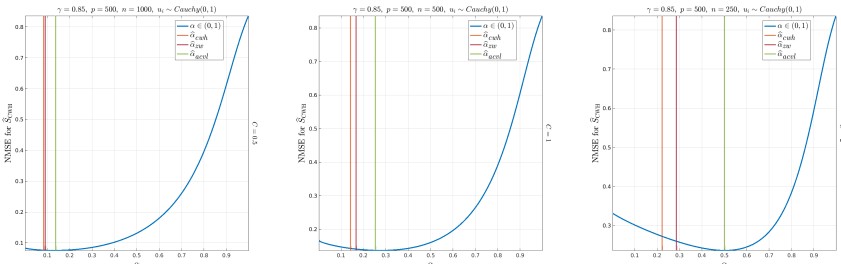

Figure 2: The solid blue line shows the NMSE between the population matrix $\mathbf{S}$ and the scatter matrix $\widehat{\mathbf{S}}$ estimated using SBP's RFPI algorithm for $\alpha \in (0,1)$ and $p = 500$, in three different settings: $p < n$ (left), $p = n$ (middle), and $p > n$ (right). The orange, red, and green solid vertical lines indicate the values for $\widehat{\alpha}_{cwh}$, $\widehat{\alpha}_{zw}$, and $\widehat{\alpha}_{acvl}$ obtained using the methods in [7, Eq. 13], [45, Eq. 12], and the ACVL method, respectively.

That is, we plugin the *Regularized* TME $\widehat{\mathbf{S}}(\bar{\alpha}; \mathcal{X}_n) \in \mathbb{S}_p^+$ from (15) into equation (18) to obtain the new weights $\widetilde{v}_j$, for $j \neq i$, $j = 1, \ldots, n$; then use the new weights $\widetilde{v}_j$ to obtain the new estimate $\widetilde{\mathbf{S}}(\bar{\alpha}; \mathcal{X}_{n\backslash i})$ in (19). Using this approximation, and for a fixed $\bar{\alpha} \in (0,1)$, computing $\widetilde{\mathbf{S}}(\bar{\alpha}; \mathcal{X}_{n\backslash i})$ does not require invoking the RFPI algorithm for each $i = 1, \ldots, n$. Instead, the RFPI algorithm will be invoked once to compute $\widehat{\mathbf{S}}(\bar{\alpha}; \mathcal{X}_n)$ in (15), while $\widetilde{\mathbf{S}}(\bar{\alpha}; \mathcal{X}_{n\backslash i})$ in (19) can be directly computed for each $i = 1, \ldots, n$. Using the approximation in (19), an optimal $\alpha$ can now be computed as follows:

$$\widehat{\alpha}_{CV}^* = \underset{\alpha \in (0,1)}{\arg\min} \quad \widetilde{\mathcal{L}}_{CV}(\mathcal{X}_n, \alpha) \text{ , where} \tag{20}$$

$$\widetilde{\mathcal{L}}_{CV}(\mathcal{X}_n, \alpha) = \frac{1}{n} \sum_{i=1}^{n} \mathcal{L}(\mathbf{x}_i, \widetilde{\mathbf{S}}(\alpha; \mathcal{X}_{n\backslash i})) \text{ ,} \tag{21}$$

and $\widetilde{\mathcal{L}}_{CV}(\mathcal{X}_n, \alpha)$ is the *approximate cross-validated loss* (ACVL); and we refer to the problem in (20) as the ACVL method. For $m$ values of $\alpha$ in $(\alpha_1, \ldots, \alpha_m)$, the RFPI algorithm will now consume $O(m * T(np^2 + p^3))$ running time from the ACVL method.

## 5 Experimental Results

In this section, we evaluate the performance of the ACVL method on synthetic and real high-dimensional datasets, and compare it with other shrinkage coefficient estimation methods in the literature; in particular the methods proposed in [7], [45], and [27]. For synthetic data, and similar to other works in the literature on RTME [7, 43, 37, 40, 35, 15], we consider the Toeplitz matrix used in [3] to be the population scatter matrix $\mathbf{S}$ for the elliptical RV in (1); i.e. $\mathbf{S} = (s_{i,j}) = \gamma^{|i-j|}$, where $\gamma = \{0.1, 0.5, 0.85\}$. Note that $\mathbf{S}$ approaches a singular matrix when $\gamma \to 1$, and $\mathbf{S}$ approaches the identity matrix when $\gamma \to 0$.

The random quantities $u$ and $\mathbf{y}$ in (1) are stochastically independent. We let $\mathbf{y}_1, \ldots, \mathbf{y}_n$ be samples from a $p$-variate standard Gaussian distribution $N(\mathbf{0}, \mathbf{I})$. For r.v. $u$, we consider four different choices for heavy-tailed distributions: (*i*) $u_i = 1$, which makes $\{\mathbf{z}_1, \ldots, \mathbf{z}_n\}$ are *i.i.d.* samples from $N(\mathbf{0}, \mathbf{S})$; (*ii*) $u_i = \sqrt{d/\chi_d^2}$, a Student-T distribution with degrees of freedom $d = 3$; (*iii*) $u_i = \text{Laplace}(0,1)$, a heavy-tailed distribution with finite moments; and (*iv*) $u_i = \text{Cauchy}(0,1)$, a heavy-tailed distribution with undefined moments. Note that since TME and RTME operate on the normalized samples $\mathbf{x}_i$, the scalars $u_i$'s cancel out, and the resulting plots become identical regardless of the distribution of $u_i$. For this reason, we show here only the plots for the multivariate Cauchy distribution.

The accuracy of an estimator $\widehat{\mathbf{S}}$ is measured using the normalized mean squared error (NMSE) $\|\widehat{\mathbf{S}} - \mathbf{S}\|_F^2 / \|\mathbf{S}\|_F^2$. The convergence criterion for all RFPI algorithms is $\|\widehat{\mathbf{S}} - \mathbf{S}\|_F^2 < \epsilon$, where $\epsilon = 1.0e - 9$ is the desired solution accuracy. For Figure (2), $p$ is set to 500, while $n$ is set to three different values $\{1000, 500, 250\}$ to consider three different scenarios: $p < n$, $p = n$, and $p > n$, respectively. The value of $C$ that appears on the right $y$-axis in all figures is for the ratio $p/n$.

Figure (1) compares the *Exact* CV loss to the *Approximate* CV loss developed in the previous section, for the multivariate Cauchy distribution (which has undefined moments). It can be seen that the Exact CV loss in (13) (solid blue line) and the Approximate CV loss in (21) (solid red line) are *almost identical* in all settings: $p < n$, $p = n$, $p > n$, and for all values of $\gamma = \{0.1, 0.5, 0.85\}$. This negligible difference between the Exact and Approximate CV loss supports our proposal that the latter can be leveraged to estimate a near-optimal value for the shrinkage coefficient $\alpha$. This can be confirmed by noticing that the optimal $\alpha$ estimated using the ACVL method (red square) is reasonably close to, or overlaps, the optimal $\alpha$ estimated using the Exact CVL (blue circle) in all nine settings for the multivariate Cauchy distributions. In terms of speedup, the ACVL method is at least $20\times$ faster than the Exact CVL method in all the different settings (see exact runtimes in the legends of Figure (1)).

Figure (2) compares the shrinkage coefficient estimated using the ACVL method in (20), denoted by $\widehat{\alpha}_{acvl}$, with the shrinkage coefficients estimated from the closed-form expressions in [7, Equation 13] (CWH), denoted by $\widehat{\alpha}_{cwh}$, and [45, Equation 12] (ZW), denoted by $\widehat{\alpha}_{zw}$. While the methods in [7] and [45] are faster than the ACVL method due to their closed-form expressions, it can be seen that the ACVL method provides more accurate estimates for $\alpha$ especially when $p \geq n$. Also, it can be noticed that the estimates from [7] and [45] tend to *diverge* from the optimal value as $p$ is growing greater than $n$. The tendency for methods based on asymptotic analysis and RMT results to over/under estimate the value for $\alpha$ is understandable since such methods make explicit assumptions about the data's underlying distribution. This over/under estimation of $\alpha$ leads to larger values of the NMSE as shown in Figure (2), as well as larger values for the LOOCV NLL loss as demonstrated in the following experiments.

Tables $(1 - 4)$ in Appendix A compare the LOOCV NLL loss for the scatter matrices estimated using Ledoit–Wolf (LW) estimator [27], and the RTME with shrinkage coefficients from CWH [7], ZW [45], and the ACVL method in (20). The comparison between the different estimators was carried out using four real high-dimensional datasets: (*i*) Images for the first six subjects from the Extended Yale B dataset for face recognition [14]; (*ii*) Images for the first six object categories from the test set for the CIFAR100 dataset for object recognition; (*iii*) Images for the first six object categories from the test set for the CIFAR10 dataset for object recognition; and (*iv*) Images for the first six digits' classes (0, 1, 2, 3, 4, 5) from the United States Postal Service (USPS) dataset for handwritten digits [33].

From Tables $(1 - 4)$ it can be seen that for most of the cases, scatter matrices estimated using RTME yield lower LOOCV NLL loss than the scatter matrices estimated using LW estimator. The difference in performance between both classes of estimators is primarily due to the difference in the underlying assumption on data distribution; hence, both classes derive different estimation procedures for their respective scatter matrices. While LW estimator assumes that the data are sampled from a multivariate Gaussian distribution, the class of TME and RTME assume that the data are sampled from a multivariate elliptical distribution with heavy tails (Eq. 3). The better performance for RTME suggests that the class of multivariate elliptical distributions can be a better alternative than the Gaussian distribution for modeling high-dimensional data with an (unknown) empirical distribution. In terms of shrinkage coefficients for RTME, it can be seen that the ACVL method yields lower LOOCV NLL loss than the methods in [7] and [45] for all cases in Tables $(1 - 4)$. This confirms our earlier observation that over/under estimation of the coefficient $\alpha$ leads to larger LOOCV NLL loss which, potentially, may jeopardize the performance of one or more downstream inferential tasks.

## 6   Conclusion

Robust estimation of a high-dimensional covariance matrix from empirical data is a well-known challenging task, especially when $p \geq n$. In this work, we considered RTME, an accurate and robust estimator for the scatter matrix when the data are samples from an elliptical distribution with heavy tails, and $p \geq n$. In particular, our work presented here introduces an alternative approach for estimating an optimal shrinkage coefficient $\alpha$ for RTME, focusing on both accuracy and computational efficiency. Unlike existing methods, our approach uses efficient computation and the given finite sample to estimate a near-optimal $\alpha$ for RTME. The main driver for this efficiency is the Approximate LOOCV NLL loss for the estimated scatter matrix with respect to parameter $\alpha$ (Eq. 20). As a result, the ACVL method demonstrated competitive performance in experiments with high-dimensional synthetic and real-world data. An interesting question for future work is whether the proposed approximation can be extended to other covariance matrix estimators, or more generally,

to hyper-parameters' selection for different classes of learning algorithms. Another research direction can explore further approximations for the LOOCV loss where the approximation can better exploit the specific structure of the learning algorithm; e.g. algorithms for subspace learning, and algorithms for learning mixture models.

# Appendix

## A Tables

Table 1: Comparison results for the first 6 (out of 38) classes from the Extended Yale B dataset; $n = 64$, $p = 1024$.

| ID | LW | CWH | ZW | ACVL |
|----|------|------|------|----------|
| 1 | 5677 | 5371 | 5643 | **3705** |
| 2 | 5613 | 5440 | 5598 | **3706** |
| 3 | 5768 | 5470 | 5749 | **3826** |
| 4 | 5403 | 5080 | 5362 | **3455** |
| 5 | 5824 | 5435 | 5786 | **3716** |
| 6 | 5797 | 5460 | 5761 | **3790** |

Table 2: Comparison results for the first 6 (out of 20) classes from the CIFAR100 *test set*; $n = 500$, $p = 1024$.

| Label | LW | CWH | ZW | ACVL |
|---------|------|------|------|----------|
| apple | 849 | 866 | **846** | **846** |
| aq.fish | 807 | 810 | 799 | **767** |
| baby | 968 | 984 | 967 | **932** |
| bear | 810 | 794 | 803 | **769** |
| beaver | 869 | 846 | 859 | **812** |
| bed | 1051 | 1047 | 1043 | **1008** |

Table 3: Comparison results for the first 6 (out of 10) classes from CIFAR10 *test set*; $n = 1000$, $p = 1024$.

| Label | LW | CWH | ZW | ACVL |
|-------|-----|-----|-----|---------|
| airp | 631 | 593 | 612 | **590** |
| auto | 913 | 900 | 906 | **894** |
| bird | 727 | 705 | 718 | **694** |
| cat | 773 | 757 | 772 | **755** |
| deer | 769 | 753 | 761 | **739** |
| dog | 721 | 702 | 719 | **699** |

Table 4: Comparison results for the first 6 (out of 10) classes from the USPS dataset; $p = 256$. Note that $n$ varies for each digit's class

| Label | $n$ | LW | CWH | ZW | ACVL |
|-------|------|------|------|------|----------|
| 0 | 1585 | 268 | 259 | 261 | **239** |
| 1 | 1330 | -269 | -374 | -309 | **-475** |
| 2 | 952 | 370 | 366 | 369 | **342** |
| 3 | 807 | 336 | 327 | 330 | **298** |
| 4 | 795 | 310 | 293 | 301 | **249** |
| 5 | 659 | 360 | 357 | 359 | **337** |

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
