# OpenReview forum: "Approximate Leave-One-Out Cross Validation for Robust Scatter Matrix Estimation"
_NeurIPS.cc/2025/Workshop/Reliable_ML — NeurIPS 2025 - Reliable ML Workshop_

### Official Review · Reviewer_1WXx · 2025-09-18
**Efficient and Promising Approximation for LOOCV in Regularized Tyler’s Estimation with Room for Further Analysis**

**Rating:** 7
**Confidence:** 2

**Review:**

## Summary. What the paper claims, how it does it, and the main results.
The paper proposes an approximate leave-one-out cross-validation (ACVL) scheme for Regularized Tyler’s M-estimator, which replaces the need to recompute the estimator n times by using the full-sample solution to construct closed-form leave-one-out surrogates, thereby enabling efficient and accurate selection of the shrinkage parameter. In both synthetic dataset and the real-world experiments, the proposed method achieved more efficient and accurate result compared to the baselines.

## Strengths. Novelty, rigor, empirical/theoretical quality, clarity, relevance to reliability with imperfect data.
This paper is novel, theoretically solid and also have empirical result to support its claims that the proposed ACVL is more efficient.

## Weaknesses / Limitations. Missing comparisons/ablations, unclear assumptions, proof gaps, failure modes, scope limits.
1. The proposed method use the full-sample solution as the surrogate for the leave-one-out surrogates. Intuitively, the approximation error will increase as n getting smaller which is exactly the motivation that n < p. In addition, the approximation error will also increase as the data getting more and more noisy. It seems that the author did not include the robustness analysis of the proposed method.
2. I can understand that this proposed method is more efficient than the baselines, but in the experiment, the error is even less compared to the baselines. Is it possible for the author to provide explanation on that?
3. The goal is to solve a low-dimensional (1D) optimization problem whose target is costly to evaluate. From my point of view, this seems very suitable to use Bayesian Optimization, where the cost of function evaluation is not reduces but the number of evaluation is reduced. Is it possible for the author to include this as a baseline?

## Suggestions for Authors. Specific things that would improve the paper
See weakness

## Ethics (if applicable). Note any concerns (about privacy, fairness, misuse, sensitive data use) and suggested mitigations
N/A

---

### Official Review · Reviewer_nCn9 · 2025-09-19
**Approximate Leave-One-Out Cross Validation for Robust Scatter Matrix Estimation**

**Rating:** 7
**Confidence:** 4

**Review:**

The paper was aimed at performing estimation of the scatter matrix in settings of $p > n$. In these settings, the standard technique is to perform a form of regularized maximum likelihood estimation with a shrinkage parameter $\alpha$. The goal of the paper was then to identify an efficient procedure to perform optimal $\alpha$ selection with LOOCV. The idea of the paper was to then do the estimation of this LOOCV by first fitting an estimator on the entire dataset for each choice of $\alpha$ and then using such an estimator to approximate the LOO estimators across the dataset, thereby more efficiently performing this evaluation.

The idea of the paper is well motivated and approach also seems well founded. Adding in analysis on the suboptimality or convergence guarantees for large $n$ of the approximation would be an interesting addition to consider for a fully fleshed-out paper.